# Understory Vegetation Change Following Woodland Reduction Varies by Plant Community Type and Seeding Status: A Region-Wide Assessment of Ecological Benefits and Risks

**DOI:** 10.3390/plants9091113

**Published:** 2020-08-28

**Authors:** Thomas A. Monaco, Kevin L. Gunnell

**Affiliations:** 1U.S. Department of Agriculture, Agricultural Research Service, Forage and Range Research Laboratory, Utah State University, Logan, UT 84322-6300, USA; 2Great Basin Research Center, Utah Division of Wildlife Resources, Ephraim, UT 84627, USA; kevingunnell@utah.gov

**Keywords:** conifer encroachment, large-scale restoration, seeding, ecological site potential, woodland expansion, effect-size analysis, regional assessment, vegetation analysis, functional group

## Abstract

Woodland encroachment is a global issue linked to diminished ecosystem services, prompting the need for restoration efforts. However, restoration outcomes can be highly variable, making it difficult to interpret the ecological benefits and risks associated with woodland-reduction treatments within semiarid ecosystems. We addressed this uncertainty by assessing the magnitude and direction of vegetation change over a 15-year period at 129 sagebrush (*Artemisia* spp.) sites following pinyon (*Pinus* spp.) and juniper (*Juniperus* spp.) (P–J) reduction. Pretreatment vegetation indicated strong negative relationships between P–J cover and the abundance of understory plants (i.e., perennial grass and sagebrush cover) in most situations and all three components differed significantly among planned treatment types. Thus, to avoid confounding pretreatment vegetation and treatment type, we quantified overall treatment effects and tested whether distinct response patterns would be present among three dominant plant community types that vary in edaphic properties and occur within distinct temperature/precipitation regimes using meta-analysis (effect size = *lnRR* = ln[posttreatment cover/pretreatment cover]). We also quantified how restoration seedings contributed to overall changes in key understory vegetation components. Meta-analyses indicated that while P–J reduction caused significant positive overall effects on all shrub and herbaceous components (including invasive cheatgrass [*Bromus tectorum*] and exotic annual forbs), responses were contingent on treatment- and plant community-type combinations. Restoration seedings also had strong positive effects on understory vegetation by augmenting changes in perennial grass and perennial forb components, which similarly varied by plant community type. Collectively, our results identified specific situations where broad-scale efforts to reverse woodland encroachment substantially met short-term management goals of restoring valuable ecosystem services and where P–J reduction disposed certain plant community types to ecological risks, such as increasing the probability of native species displacement and stimulating an annual grass-fire cycle. Resource managers should carefully weigh these benefits and risks and incorporate additional, appropriate treatments and/or conservation measures for the unique preconditions of a given plant community in order to minimize exotic species responses and/or enhance desirable outcomes.

## 1. Introduction

Semiarid ecosystems are currently threatened by woody plant dominance due to encroachment (i.e., spreading) and infilling (i.e., densification), heightening the need to understand how these changes impact ecosystem functioning and critical ecosystem services [1,2]. Numerous factors, including elevated atmospheric CO_2_, increased N deposition, climate shifts, reductions in fire, and changes in grazing/browsing regimes are believed to play an important role in woody plant encroachment [2]. Similar to global trends, multiple interacting factors have been attributed to coniferous tree expansion (e.g., single-leaf piñon pine (*Pinus monophyla* Torr. and Frém.), Colorado piñon pine (*P. edulis* Engelm.), Utah juniper (*Juniperus osteosperma* Torr.), and western juniper (*Juniperus occidentalis* Hook); hereafter P–J (pinyon–juniper)) into semiarid shrub-steppe ecosystems in the Intermountain Region of western North America [3,4,5]. These factors include (1) natural range expansion [6], (2) decreased fire frequency due to cessation of periodic fires after European arrival, active fire control, and the creation of fire barriers [3,7,8,9,10,11], (3) introduction of livestock grazing and heavy grazing following the arrival of Europeans that reduced fuels and altered competitive interactions between herbaceous species and trees [4,8,10,12,13,14,15], (4) favorable climatic conditions, especially during wetter and cooler conditions between 1900 and 1950 [16,17,18,19,20], (5) afforestation following prior woodland reduction and harvesting [21], and (6) woodland recovery from megadroughts in late 1500s [6,10]. Prior to European settlement of the western U.S., P–J woodlands occurred on fewer than 3 million ha, but estimates indicate distribution across foothills, mesas and plateaus, and low mountain woodlands now occupy an area as high as 50 million ha in western U.S. [22,23], more than 18 million ha in the Intermountain West [17,22,24,25], and between 4 and 6 million ha in Utah alone (i.e., more than 25% of its land area) [23,25,26,27]. Current estimates also indicate that P–J woodlands have increased within the range of 125–625% since 1860 due to encroachment into shrub-steppe ecosystems that did not previously support trees and infilling within shrub-steppe woodlands [12,14,16,19,20,28,29,30]; yet patterns have not been uniform throughout the Intermountain Region and the relative importance of potential factors causing P–J expansion is largely unknown for most locations in this region [3].

While woody plant encroachment does not universally degrade ecosystems [31], in the Intermountain Region of western North America it is a primary conservation concern due to negative impacts on sagebrush (e.g., black sagebrush, *Artemisia nova* A. Nelson and big sagebrush, *A. tridentata* spp. Nutt.)-dominated semiarid shrubland and shrub-steppe ecosystems [4,7,8,14,16,20,32,33,34,35]. Encroachment of P–J in this region has been linked specifically to sharp reductions in herbaceous understory vegetation and species diversity [30,36,37,38,39,40,41,42,43,44,45], increases in flammable exotic annual grasses and the risk of creating an annual grass-fire cycle [32,33,46,47], soil instability, soil erosion, and reduced hydrological functioning [48,49,50,51,52,53]. The reduction and/or absence of desirable herbaceous vegetation on encroached sites has also led to inadequate seed banks to allow natural regeneration after the application of P–J reduction treatments [54,55,56], but see [57]. Furthermore, degraded understory vegetation associated with increasing tree density diminishes habitat suitability for wildlife species, including regionally important mule deer (*Odocoileus hemionus*) [42,58] and greater sage-grouse (*Centrocercus urophasianus*) [59,60,61,62,63,64]. Hazardous woody fuel build-up is also threatening sagebrush communities due to the extreme risk of intensive wildfires [12,19,65]. Given these conservation concerns and ecological impacts of woodland encroachment, proactive P–J reduction and restoration seeding are viewed as ways to improve the capacity of shrublands and shrub-steppe plant communities to support greater ecosystem services, including expediting the recovery of understory native shrub and herbaceous species [66,67], and creating suitable habitats for imperiled avian species [61,68]. Furthermore, proactive management associated with mechanical P–J reduction increases water accumulation in winter, infiltration rates following precipitation events, and soil water availability in spring [15,69,70,71,72,73,74], thus altering key ecological processes necessary to enhance understory vegetation [67,75,76]. However, the conservation benefits of such treatments have not been consistently realized [1,77,78,79,80], the efficacy of this management strategy has been highly variable, and the longevity of removal/reduction treatments often do not exceed 10 years [2,78]. Consequently, there is critical need for empirical assessments of large-scale restoration projects at the regional scale to uncover patterns in posttreatment vegetation dynamics, enhance our ability to choose the most appropriate site-specific treatments for future restoration [80,81,82,83], and inform the public on how restoration activities are achieving management goals [84,85,86,87,88].

Although the factors responsible for idiosyncratic restoration outcomes in woodland ecosystems are not fully understood [67,78,79,89,90], evidence indicates that pretreatment vegetation and unique biophysical conditions prevalent within distinct plant communities are important determinants of understory recovery [91,92,93,94] and habitat suitability for ground-nesting birds [95,96,97]. For example, the recovery of understory vegetation following P–J removal depends on both pretreatment levels of woodland encroachment [4,12,49,98] and abundance of native vegetation [39,79,91,99,100,101,102], which are often inversely related. This inverse relationship indicates that strong competitive interactions for soil resources are responsible for the contingence between pretreatment tree canopy cover and/or density in P–J woodlands and posttreatment herbage production [30,39,45,56,57,103,104,105]. For example, because rooting zones of trees can overlap substantially [106] and strongly compete with understory vegetation for limiting resources [107,108], P–J reduction is expected to liberate soil resources necessary for the understory recovery [76,94]. Thus, sites with greater pretreatment understory abundances of perennial grasses and native shrubs are expected to have higher recovery potential compared to sites with advanced phases of woodland development and severely degraded understory vegetation [94,109]. Sites with advanced woodland development are also more prone to invasive grass increases after P–J removal than less-developed woodland sites [39,91,110], but when pretreatment native vegetation contains a high abundance of perennial grasses, understory recovery can preclude posttreatment dominance of exotic annual species by competing for resources made available after P–J removal [111,112,113,114,115] as well as influence posttreatment native shrub abundance [90,113].

The paramount influence of pretreatment vegetation on posttreatment understory recovery may in fact muddle the interpretation of restoration outcomes in woodland ecosystems because the choice of treatment type is usually based on pragmatic and/or workable features of treatment applications, which can confound a clear interpretation of the influences of treatment type and pretreatment vegetation on posttreatment responses. For example, in the absence of fire to naturally regulate woodland encroachment [12,19,116], numerous mechanical treatments (i.e., chaining, mastication; e.g., shredding and dispersing mulch, and cutting) have been developed to function as fire surrogates for fuel reduction, watershed improvement, and to restore understory vegetation components [67,76,90,117,118,119]. However, the suitability of each treatment depends on pretreatment vegetation and ecological site characteristics [5,32], including the amount of understory herbaceous and shrub species and the severity of P–J encroachment [117,120]. Chaining is typically applied to sites with larger-diameter trees, higher tree cover, and degraded understory vegetation [121,122] and creates greater ground disturbance than mastication and cutting, which can in turn increase the density of invasive annual species [89,115,123,124]. Thus, chained sites are nearly always seeded and inherent soil disturbance associated with chaining is considered necessary to alter seed bed conditions and increase establishment of seeded species [95,125,126,127,128,129,130,131,132,133,134,135]. Mastication is also suitable for sites characterized by later stages of woodland development but is followed by seeding only if pretreatment understory conditions are degraded [40,60,89]. Mastication is also unique compared to the other treatments due to its production and dispersal of mulched residue that reduces bare ground, erosion and runoff [136,137,138,139,140], increases water infiltration rates, and reduces sediment yields relative to areas lacking the masticated residue [137]. Studies also indicate that this residue can potentially reduce seedling emergence of seeded species [132,141] and the tracked vehicles used to apply this treatment can decrease soil aggregate stability [137,142]. In contrast to chaining and mastication, cutting maintains understory shrub and herbaceous cover with minimal ground disturbance [91,102,124,139] and is most appropriate for sites with low tree density that do not require seeding [67,69,139,143]. Consequently, due to pretreatment vegetation conditions and disturbance regimes intrinsic to each treatment type, treatment types are expected to yield variable restoration outcomes [144,145], but it remains difficult to extrapolate restoration outcomes from disparate studies to other situations because treatment efficacy and pretreatment conditions are not mutually independent.

As P–J encroachment has occurred over a diverse range of topographic, climatic, and edaphic conditions in the Intermountain West [24,26,146,147,148], categorical plant community classifications that incorporate biophysical-site properties and vegetation-recovery potentials offer a practical platform to make impartial comparisons among treatment alternatives and decipher restoration outcomes [90,149]. Thus, constraining the direct comparison of treatment alternatives within plant community types can partially address the uncertainties created by idiosyncratic restoration outcomes in woodland ecosystems [78] and reveal the site-specific conditions where restoration may be most successful [94,109,150,151,152]. For example, plant communities encroached by P–J species are commonly classified by sagebrush taxa [147,153,154,155,156] that dominate within distinct soil temperature/soil moisture regimes. At one extreme, higher elevation mountain big sagebrush communities are characterized by cool/moist (i.e., frigid/xeric) precipitation/temperature regimes, receive higher average annual precipitation, and have higher soil water holding capacity compared to Wyoming big sagebrush and black communities that dominate warm/dry (mesic/aridic) regimes at lower elevations and on soils with lower water holding capacity [157,158,159,160]. Black sagebrush communities also dominate on gravelly soils with clay-textured or calcified subsoil horizons (i.e., caliche) that create shallow rooting depths and poor water infiltration compared to communities dominated by big sagebrush species [161,162]. Lower elevation black- and Wyoming-big sagebrush communities experiencing P–J encroachment are also more susceptible to annual grass invasion compared to mountain big sagebrush communities in the Intermountain Region [163,164], which leads to key differences in resistance to exotic plant invasion and resilience following disturbance and environmental stress [109,165,166,167]. These differences underpin the capacity of plant community types to serve as effective environmental surrogates [109,165] and emphasize the need to assess restoration outcomes of alternative P–J reduction treatments within plant community types in order to enhance site-specific management recommendations [165,168].

In this study, we assessed vegetation change over a 15-year period following landscape-scale P–J reduction applied at 129 woodland sites in Utah, USA. Our overarching objective was to assess both restoration benefits (i.e., positive effects on understory herbaceous-perennial and shrub components) and ecological risks (i.e., positive changes in exotic annual species) associated with this effort to reverse woody plant encroachment. To do this, we evaluated the relationships among pretreatment vegetation components and calculated effect-sizes for eight vegetation and soil surface variables to uncover the contingence of posttreatment vegetation change on plant community type for three common mechanical P–J reduction treatments using meta-analysis; an analytical approach particularly suitable to study different disturbance types applied across highly variable conditions [85,144]. Meta-analysis was also used to assess how restoration seedings contributed to overall changes in understory vegetation components within plant community types. Due to strong competitive interactions between P–J species and understory vegetation, we expected that increased soil resources following P–J reduction treatments would ubiquitously lead to significant positive changes in understory vegetation. However, because treatment types with higher disturbance intensity are routinely applied to sites with more advanced pretreatment woodland development, we expected that the magnitude of change for herbaceous and shrub components would be greater for chaining and mastication compared to the lower intensity cutting treatment. We also expected positive changes would be larger for understory vegetation in mountain big sagebrush communities that typically exhibit greater resilience to disturbance than other sagebrush communities (e.g., [94,157,169]). Reconciling these expectations will better inform ongoing conservation efforts to offset woody plant encroachment in semiarid ecosystems and refine the development of management guidelines that incorporate site-specific criteria when planning and executing restoration projects.

## 2. Results

### 2.1. Pre-Treatment Vegetation

Pretreatment differences among mechanical treatments and plant community types were found for six of the eight response variables (Appendix A). Significant differences included higher P–J cover for chaining and mastication compared to cutting and variation in the amounts of both sagebrush and perennial grasses ranging from low values for chaining, intermediate values for mastication, and high values for cutting (Figure 1A). In contrast, differences among plant communities included lower annual grass cover and higher cryptogam cover, in black sagebrush communities compared to big sagebrush communities (Figure 1B). Perennial forb cover was significantly greater for mountain big sagebrush than Wyoming big sagebrush communities, while values were intermediate for black sagebrush communities. Inverse relationships between pretreatment P–J cover and both sagebrush and perennial grass cover were significant for all plant community types with the exception of sagebrush cover in black sagebrush communities (Figure 2A,B). Lastly, pretreatment cover values (mean ± SE) were significantly greater on unseeded sites than seeded sites for sagebrush (5.6 ± 0.9 vs. 2.7 ± 0.8, respectively; *t* = 3.1; *p* = 0.002), perennial grass (9.8 ± 1.0 vs. 1.8 ± 0.5, respectively; *t* = 7.3; *p* < 0.001), and perennial forb (0.9 ± 0.2 vs. 0.4 ± 0.1, respectively; *t* = 2.7; *p* = 0.008).

### 2.2. Plant Community Responses to P–J Reduction

The overall effects of P–J reduction were significant for all eight response variables; effects were positive for sagebrush, all herbaceous components, and soil surface variables but negative for P–J and bare ground (Figure 3, Figure 4 and Figure 5). Treatment effects (i.e., communities pooled) were also more positive for chaining and mastication compared to cutting for herbaceous vegetation, but plant communities did not generally respond differently within a treatment type except for annual grass (Figure 4C) and cryptogam (Figure 5B), which were more positive for black sagebrush communities within the chaining treatment.

Although plant community types differed for only two of the eight variables, the significance of effect sizes (i.e., *H_o_*: μ = 0; α = 0.05) varied among plant communities in many instances. For example, greater resilience of mountain big sagebrush communities was evident from significant positive effect sizes for mountain big sagebrush communities but not the other community types for sagebrush cover (i.e., chaining treatment; Figure 2B), perennial grass cover (i.e., cutting treatment; Figure 3A), and a significant negative effect size for bare ground (i.e., cutting treatment; Figure 5A). In contrast, a non-significant effect size for bare ground within the chaining treatment for Wyoming big sagebrush communities indicated that vegetation recovery did not result in parallel changes in bare ground as was observed in the other community types (Figure 5A). Non-significant effects for perennial and annual forb cover within black sagebrush communities in the chaining treatment, where positive effects on annual grasses were most pronounced, also differs from responses seen in the two big sagebrush communities (Figure 4B,D). However, unlike chaining, the effect size for annual grass cover was not significantly different than zero for black sagebrush communities in the mastication treatment yet chaining had a significant effect on annual grass cover in both big sagebrush community types (Figure 4C).

### 2.3. Influence of Seeding on Post-Treatment Understory Recovery

Overall effect sizes for seeding were significant for perennial grass and perennial forb, but not sagebrush (Figure 6). Seeding also had significant, positive effects on perennial grass and perennial forb in many plant community–treatment combinations, but significant differences between seeded and unseeded sites were observed only at mountain big sagebrush sites for perennial forb (regardless of treatment) and for perennial grass in the mastication treatment. Lastly, overall annual grass responses were not significantly different between seeded and unseeded sites, nor were differences found for any of the treatment–plant community combinations (data not shown).

## 3. Discussion

### 3.1. Overall Effects of P–J Reduction and Seeding were Dependent on Disturbance Intensity and Pre-Treatment Vegetation

The overall effects of reversing woody plant encroachment through mechanical P–J reduction in our assessment revealed positive changes in all understory vegetation components and supported the expectation that higher disturbance associated with chaining and mastication would lead to more dramatic changes in herbaceous components relative to the cutting treatment. Thus, although high disturbance intensity can greatly alter soil surface attributes, injure non-target plants, and disrupt species recruitment [111,168,170,171], our results illustrated that sites treated with chaining and mastication stood to gain the most from P–J reduction (i.e., based on pretreatment conditions; Figure 1A) and experienced changes in understory conditions yielding both beneficial ecological services and undesirable ecological risks. In contrast, it was surprising that comprehensive positive changes in understory vegetation were not more prominent in mountain big sagebrush communities that typically exhibit greater resilience to disturbance than other sagebrush communities [94,109]. Instead, differences among plant communities were rarely observed within treatment types (i.e., annual grass and cryptogam under chaining; Figure 4C and Figure 5B, respectively); however, both significant and non-significant effect sizes were observed for all response variables except pinyon–juniper when comparing plant communities within treatment types. These nuanced results, thus, partially support the expected resilience of plant community types and highlight specific situations where caution is warranted in order to avoid unintended consequences associated with P–J reduction. In addition, while we observed dramatic overall effects of seeding on perennial grass and forb components in support of the widely held expectation that seeding contributes to understory recovery [66,142,165,172], these increases must be examined from the perspective of pretreatment conditions that dictated whether seeding was necessary as well as the treatment–plant community backdrop in order to fully understand and appraise situations where seeding can yield the most benefits to understory recovery.

### 3.2. Why Were Understory Responses More Pronounced for Chaining and Mastication Than Cutting?

While our results agree with previous studies indicating that pretreatment levels of encroachment and the abundance of residual perennial vegetation are key factors associated with variation in treatment outcomes [4,12,39,49,91,98,173], we stress that less-pronounced overall changes in herbaceous perennial vegetation within the cutting treatment should not be viewed as an unfavorable treatment outcome [89,103]. Instead, this pattern reflects the fact that cutting sites were in earlier phases of woodland development that had inherently greater pretreatment residual vegetation to support understory recovery (i.e., lower pretreatment P–J cover and higher pretreatment sagebrush perennial grass cover; Figure 1A; [67,91,174]). Our results also emphasize that although higher initial cover of perennial vegetation equated to lower overall posttreatment change for cut sites, positive responses in perennial grass and cryptogam suggest that increases in soil water and nutrients that typically accompany P–J reduction likely triggered these significant responses [39,72]. These changes in understory vegetation and soil surface cover probably contributed to the competitive effects of perennial grasses and the absence of positive changes to undesirable annual grass and forb components (i.e., Figure 4C,D) that typically increase following P–J reduction [125,175] as was the case for chaining and mastication. It is also feasible that greater residual perennial grass plants might have suppressed perennial forbs on cutting sites as reported in other studies when perennial grasses begin to dominate over time [71,110,126,176]. This interpretation is based on the understanding that herbaceous species typically exhibit overlapping resource use, perennial grasses are effective competititors, and that cryptogams can potentially prevent annual grass establishment [177,178,179].

In contrast to cutting, inherently low pretreatment perennial cover grass in the chaining and mastication treatments magnified the capacity of annual grasses to proliferate following P–J reduction [89,98]. For example, chaining directly disturbs the soil surface, creating favorable safe-site microenvironments for seedling emergence and establishment [180,181]. While this mechanism explains the robust changes associated with chaining in understory vegetation, including the only significant positive change in sagebrush (i.e., at mountain big sagebrush sites), it had a similar effect on annual grasses, particularly at black and mountain big sagebrush sites [79,182,183,184]. Mastication is also known to promote annual grass establishment, more so than perennial grasses ([98,132,141,185,186,187], but see [188]) because the production and distribution of mulch favors annual grass growth by reducing soil temperature, increasing soil moisture, and elevating inorganic nitrogen supply to plants [39,73,132,189,190]. These results indicate that annual grasses will likely proliferate in the short-term even when perennial grasses increase following P–J reduction [116,142,187]. However, because perennial grasses are known to effectively suppress annual grasses (i.e., [110,175,191]), the expectation is that steady increases in perennial grass cover will diminish this threat over time with proper posttreatment management [67,118,167,191,192]. Consequently, extra effort during the posttreatment period (i.e., [5,167,192]) will be essential to enhance the capacity of understory herbaceous vegetation to recover and mitigate the risk of stimulating an annual grass-fire cycle (i.e., [39,116,142,193]). In addition, because livestock grazing is a key factor in the expansion of P–J through its direct influence on both perennial grass and shrub cover, it may be necessary to adjust management plans to reduce the speed of P–J recovery on treated sites [5,194,195].

### 3.3. What Ecological Processes Were Responsible for the Differences in Understory Resilience and Annual Grass Response among Plant Community Types?

Although sagebrush plant community types are known to vary widely in environmental and topo-edaphic characteristics [123,126,154,167,192], we found only marginal support for the expectation of more pronounced understory vegetation responses in mountain big sagebrush (i.e., cool/moist temperature/moisture regimes; moderately deep, loamy to clay loam soils; >300 mm annual precipitation; [94,157,169]) compared to Wyoming big sagebrush (i.e., warm/dry; moderately deep, loamy soils; 200–300 mm annual precipitation) or black sagebrush plant communities (i.e., warm/dry; shallow, stony, calcareous soils; <300 mm annual precipitation) [164,192,196]. Instead, our results indicated strong positive perennial grass and forb recovery after P–J reduction in the chaining and mastication treatments and similar levels of resilience among plant communities (Figure 4A,B). It was also surprising that the positive changes in perennial grass and forb seen in black and Wyoming big sagebrush sites matched those of mountain big sagebrush sites within the chaining and mastication treatments. We speculate that the unexpected positive responses for black and Wyoming big sagebrush communities might have stemmed from posttreatment increases in soil moisture (i.e., [69,72,182]) causing greater net increases in resource availability on these warm/dry sites than on cool/moist mountain big sagebrush communities that typically receive greater annual precipitation because of their distribution at higher elevations. This interpretation is also supported by the fact that lifeforms known to be effective indicators of resource availability (i.e., cryptogams and annual grasses [197,198,199,200]), experienced greater changes from chaining in the black sagebrush compared to mountain big sagebrush sites. Similarly, the positive changes in cryptogams on drier black and Wyoming sites became statistically significant in response to mastication and cutting, whereas changes were not significant on mountain big sagebrush sites. Our explanation for the unexpected resilience on the drier sites supports broadscale studies illustrating that favorable changes in site ecohydrology are more pronounced in P–J woodlands that experience greater increases in understory vegetation, which in turn reduces runoff and increases water infiltration [76,201,202].

Variation in annual grass response among plant community types in the chaining treatment were also an unexpected result of our assessment. Surprisingly, greater positive annual grass response for black sagebrush sites where pretreatment annual grass cover was lowest was opposite to the response observed in Wyoming big sagebrush communities. As significant positive changes in perennial grasses were observed during the posttreatment period for all plant community types, this pattern was probably a consequence of neither perennial nor annual forbs capitalizing on posttreatment increases in soil water and nutrients that accompany P–J reduction [44,72,203], which allowed greater positive changes for annual grasses in black compared to Wyoming big sagebrush plant communities. In contrast, competitive interactions among understory vegetation for soil resources cannot explain the positive annual grass responses we observed at higher elevation in mountain big sagebrush plant communities that are expected to have higher perennial plant productivity (i.e., cool/moist temperature/precipitation regimes) and resistance to exotic plant invasion than warm/dry sagebrush sites [44,94,169,172]. These results indicate that biotic resistance to annual grass expansion can be overridden when annual grasses are present prior to applying P–J reduction treatments, even when all perennial understory components simultaneously experience strong positive changes, and potentially provide competition for soil resources [136,187,204,205]. This conclusion also applies to mastication sites where annual grass responses were generally positive even though the other herbaceous understory components exhibited strong positive change. Thus, analogous to our previous recommendations to mitigate annual grasses after P–J reduction (i.e., Section 3.2), posttreatment management plans should routinely include interventions that target areas most susceptible to annual grass expansion. For example, while successional models recognize the risk of annual grass and annual forb expansion in the first few years after P–J reduction [5,110,206,207], broadcast herbicide applications of hotter/drier areas and/or spot herbicide applications of smaller microsites previously occupied by trees could be used during this critical period to target areas where annual species flourish due to the absence of perennial grass competition [208,209,210].

### 3.4. When Is Seeding Essential for Understory Recovery Following P–J Reduction?

Seeding after juniper removal is generally considered essential to provide effective suppression of exotic annual grasses through competition for spring soil moisture [66,72,126,130] and reestablish understory herbaceous vegetation [118,125,126,131,211], especially when native species seed reserves are depauperate [66,115,126,212,213]. However, these benefits are often assumed (i.e., the majority of mechanically treated P–J woodlands were also seeded [25,126,214,215]), and few studies have specifically isolated the benefits of P–J reduction alone versus the combined application of P–J reduction and seeding [40,89,193]. While responses between unseeded and seeded sites did not vary for annual grass as illustrated in a recent study [89], the variable herbaceous perennial responses we observed among plant communities and treatments indicate situations where seeding is most essential to achieve restoration goals.

As pretreatment abundance of perennial understory vegetation typically dictates whether sites are seeded (e.g., Figure 1; [67]), we were not surprised that the overall changes in perennial grass and forb components were more pronounced for seeded compared to unseeded sites and for mastication compared to cutting treatments. It was also not surprising that these benefits were more evident at cool, high elevation mountain big sagebrush sites that receive greater precipitation compared to warm, low elevation Wyoming big sagebrush sites ([151,172], but see [165]). In fact, only mountain big sagebrush sites treated with mastication showed consistently greater positive changes to both herbaceous components for seeded compared to unseeded sites. These results agree with previous studies reporting greater increases for mastication than cutting [142] and enhanced seedling establishment in mastication treatments where mulch deposition ameliorates seedbed temperature and increases surface soil water availability and inorganic soil nitrogen content [73,98,132,189,190]. Our results also indicated that the majority of responses for perennial grass and forb components became significant only for seeded sites, suggesting that seeding might be necessary more often than typically prescribed [89]. In contrast, the absence of perennial forb response at both seeded and unseeded Wyoming big sagebrush sites treated with cutting illustrates that seeding may not be necessary in this situation because cutting sites typically have higher pretreatment abundance of perennial understory vegetation and lower exotic annual weeds compared to sites receiving mastication [72,132,187,189].

Although sagebrush recovery was not deterred by P–J treatments, there were no instances across community and/or treatment types where seeding enhanced sagebrush recovery. These results suggest that factors other than seed limitation (e.g., [66,115,126]) may be responsible for poor shrub recovery. For example, sagebrush establishment is known to be both episodic [216,217] and strongly sensitive to low spring precipitation, which could have hampered seed germination and seedling establishment, even when seeds were available (e.g., [172,218]). In addition, high utilization of treated areas by big game ungulates can limit sagebrush recovery [89,219,220]. For example, our capacity to detect changes associated with seeding [165] was likely reduced by mule deer, which preferentially browse sagebrush on thinned areas to meet their winter dietary needs [221]. Lastly, perennial and annual grass seedlings strongly compete with sagebrush seedlings and can greatly reduce nutrient acquisition, growth, and survival of sagebrush seedlings [178,222,223]. In particular, rapid establishment of seeded perennial grasses can hinder the establishment of new sagebrush seedlings and slow sagebrush recovery on restoration sites [151,165,224,225].

### 3.5. Conclusions

Our results illustrated strong associations among pretreatment vegetation components, a critical factor underscoring the pressing need to reverse woody plant encroachment across this semiarid ecosystem. Furthermore, we showed that recognizing intrinsic differences in pretreatment vegetation among treatment and community types is critical to disentangle confounding factors and interpret posttreatment response to P–J reduction. We found strong and consistent changes in bare ground as well as robust changes in understory vegetation components. As expected, variation among plant community types was larger for higher-disturbance chaining and mastication treatments compared to lower-impact cutting treatments, but understory recovery was surprisingly similar among plant community types that experience highly variable temperature/precipitation regimes. However, positive changes in perennial understory components did not effectively diminish annual grass responses, which appeared to be an inherent ecological risk of modifying soil surface conditions. We also found strong positive effects of seeding on desirable herbaceous recovery for the majority of plant community–treatment combinations, suggesting that seeding is an essential determinant of whether the management objectives of reversing woodland encroachment are met or not. Finally, our results confirm that P–J reduction does not negatively impact shrub abundance [226]. Collectively, this assessment contributes to an improved understanding of restoration outcomes following landscape-level treatments to reduce woody plant encroachment and provides empirical evidence regarding how such treatments provision ecosystem services (i.e., livestock forage and wildlife habitat) via increases in understory herbaceous cover [2,227,228] and regulate additional ecosystem services (i.e., erosion control) through reductions in bare ground and increases in cryptogam cover [227,229,230].

## 4. Materials and Methods

### 4.1. Project Sites

To evaluate changes in vegetation and ground surface variables, we used data obtained from Utah Watershed Restoration Initiative (UWRI) project sites that were treated for P–J reduction between 1999 and 2016 (UWRI; https://wri.utah.gov/wri/). Project sites were primarily distributed across Utah within three Ecoregions, namely, Central Basin and Range, Wasatch and Uinta Mountain, and Colorado Plateau [231,232] and ranged in elevation between 1600 and 2454 m, with a mean elevation of 1995 m (Figure 7). Soil textures were generally classified as loam with derivations of sandy, clay, sandy clay, and silt loam. After considering all possible project sites, we selected a subset of 129 sites that met the following criteria: (1) mechanical treatments were applied to increase the abundance of big sagebrush and herbaceous species and enhance the wildlife habitat for mule deer (*Odocoileus hemionus*), Rocky Mountain elk (*Cervus canadensis*), and/or greater sage-grouse (*Centrocercus urophasianus*); (2) pretreatment plant communities could be defined by dominant shrub species (i.e., black sagebrush (*Artemisia nova*), Wyoming big sagebrush (*A. tridentata* ssp. *wyomingensis*), or mountain big sagebrush (*A. tridentata* ssp. *vaseyana*)); (3) one of three prevalent P–J reduction treatments were applied (i.e., chaining, mastication, cutting); (4) both pre- and posttreatment data were available for analysis.

Plant community types within this region differ in many factors, including soils, elevation, and biophysical indicators [233] that have been related to post-disturbance ecosystem resilience and resistance to exotic plant invasions [109,164,167,169]. For example, *A. tridentata* ssp. *vaseyana* dominates higher elevation montane areas with cold–moist temperature/precipitation regimes, greater primary productivity, and higher overall resistance and resilience capacity compared to the other two plant community types [167,169]. In contrast *A. tridentata* ssp. *wyomingensis* and *A. nova* typically occur at lower elevations with warm–dry temperature/precipitation regimes and notably lower resistance and resilience tendencies [109,158,159,167,169], with the former being more common at higher elevation moister soils and the latter dominating where shallow soils are underlain by a distinct petrocalcic (caliche) layer [157,159].

### 4.2. Treatments and Seeding

All treatments to reduce P–J abundance on project sites were applied with mechanical devices as part of the Utah Watershed Restoration Initiative (UWRI) (https://wri.utah.gov/wri/). The choice of treatment applied to each site was primarily determined by pretreatment woodland developmental phases that broadly differ in P–J cover (i.e., Phase I < 10%, Phase II = 10–30%, and Phase III > 30%; [4,12,49]); chaining treatments were accomplished by pulling a segment of a naval surplus anchor chain between two bulldozers. The chains were fashioned into an Ely chain by welding short lengths of iron across each link [180]. This treatment uproots trees, scarifies the soil surface, and creates a seedbed for broadcast seeding. To increase efficacy of uprooting trees on sites dominated by live trees (e.g., not killed by previous wildfire), the chain was pulled in two opposite directions [129]. Mastication involved using a rubber-tired or tracked industrial tractor affixed with a rotary cutter that shredded tree biomass to ground level and distributed debris in patches without creating much ground disturbance, other than soil compaction due to tractor treads [137]. Cutting was accomplished with chainsaws by felling individual trees and distributing slash haphazardly on the landscape. In general, UWRI projects apply the cutting (i.e., lop-and-scatter) treatments for plant communities classified as late Phase I to early Phase II, mastication for Phase I to Phase III communities, and chaining for late Phase II to Phase III communities. As cutting treatments typically occur in the early encroachment stages (Phase I), the understory shrub and herbaceous components of the communities are often still relatively intact; whereas tree mastication and chaining treatments typically occur in later encroachment stages (Phase II and Phase III), where the understory shrub and herbaceous components of the plant communities have been depleted to varying degrees. As such, cutting treatments seldom receive supplemental seeding, while mastication treatments receive supplemental seeding when deemed necessary, and chaining treatments receive supplemental seeding the majority of the time.

Supplemental aerial or broadcast seeding of grasses and forbs typically occurs between the first and second pass of the chain to cover seed [126]. In contrast, seeding of shrub and smaller seeded species typically occurs following the second pass. Some larger seeded shrub species such as bitterbrush (*Purshia tridentata* [Pursh] DC) and fourwing saltbush (*Atriplex canescens* [Pursh] Nutt.) may be applied using dribbler units attached to the dozer or tractor performing the treatment and pressed into the seedbed with the tire or track action or seeded with small drills following treatment. Supplemental seeding was selectively applied to certain treatment sites when land managers determined there was not an adequate seed source present to reestablish desirable understory species. Seed mixes for each study site varied, but generally consisted of native and introduced grass, forb, and shrub species with emphasis placed on establishing understory vegetation to rapidly stabilize the soil surface from erosion and provide a competitive matrix to minimize exotic annual grass invasion. Species mixes were generally broadcast-seeded aerially using either a fixed-wing airplane or helicopter, with some variation based on species and treatment method (e.g., shrub species applied by tractors doing the reduction treatments).

### 4.3. Vegetation and Ground Surface Sampling

Data collection occurred from 1997 to 2016, both prior to applying P–J reduction treatments and in subsequent years (typically every three to five years). In addition, roughly half of the project sites were monitored two- or three-times posttreatment. Sampling was conducted by establishing a 152.4-m baseline transect within treatment areas. Along the baseline transect, five 30.5 m belts were placed perpendicular on the 15.2 m mark of each belt at predetermined meter marks (3.4 m, 40.8 m, 78.9 m, 113.0 m, and 150.9 m). A steel stake was placed at the beginning of each belt to ensure consistent placement of future sampling locations. Along each of the five belts, 20, 25 × 25 cm quadrats were placed at 1.5 m intervals to measure canopy cover for herbaceous species and ground surface variables.

Canopy cover for herbaceous species was determined using an ocular cover estimation procedure using seven Daubenmire cover classes within the quadrats [234,235]. The seven cover classes were (1) 0.01–1%, (2) 1.1–5%, (3) 5.1–25%, (4) 25.1–50%, (5) 50.1–75%, (6) 75.1–95%, and (7) 95.1–100%. Similarly, with the quadrat frame on the soil surface, basal cover was also estimated for cryptogam, litter, and bare ground. It is important to note that while cryptogams encompass a broad range of lifeforms in woodland ecosystems (i.e., mosses, algae, lichens, and liverworts [236]), this category primarily consisted of bryophytes (mosses) as opposed to biological cryptogamic or microphytic crusts [237]. To determine basal and canopy cover for each belt, the midpoint for each cover class value was summed and divided by the number of sampling quadrats (i.e., 20). The five belts were used to determine mean and standard error cover percentage and for a given site. Cover of mature big sagebrush and P–J were estimated using the canopy line-intercept method [234,238]. Cover percentages were calculated by dividing the total length along each belt covered by a particular species of tree or shrub by the total length of the belt. Using individual species data, vegetation was grouped into the different categories according to functional group designation: P–J, sagebrush (i.e., *Artemisia* spp.), perennial grass, perennial forb, annual grass, and annual forb.

### 4.4. Data Analysis

To assess pretreatment vegetation, the Shapiro–Wilk Goodness-of-Fit Test was applied to all response variables and transformations to improve normality were applied based on Akaike Information Criterion values. Analysis of variance (ANOVA) was used to assess the effects of P–J reduction treatment and plant community type on pretreatment vegetation and ground cover variables (α = 0.05). For significant factors, differences among means were determined with Tukey (HSD) tests (α = 0.05). Variables that could not be transformed to meet ANOVA assumptions were analyzed for main-, but not interaction-effects with non-parametric Kruskal–Wallis tests. For significant factors, differences among means were determined with non-parametric Wilcoxon tests of each pair (α = 0.05). In addition, the associations between P–J and sagebrush and between P–J and perennial grasses were assessed using linear regression analysis in JMP ver. 14 (SAS Institute Corp. Cary, IN, USA). The significance of relationships was determined separately for each plant community type (α = 0.05). Lastly, pretreatment differences between unseeded and seeded sites for perennial grass, perennial forb, and shrub cover were analyzed using unpaired (i.e., independent samples) 2-sample Student’s *t*-test (α = 0.05).

Due to the limitations of our study design (i.e., variable treatment years, monitoring years and inequality of sites monitored each year) we quantified changes in vegetation and surface variables with standardized effect-size metrics. Effect sizes were calculated using the metafor package [239] for R (www.r-project.org) as the natural log of the ratio between post- and pretreatment (ln[post/pre] = *lnRR*) for each project site (*n* = 5) [240,241]. This approach was necessary because control areas were not available for analysis, as is typical in woody plant reduction studies. Depending on the elapsed time since treatments were applied, more than one effect size was calculated for each study site in most cases. Meta-analysis is considered an ideal method to analyze effect sizes and synthesize the outcomes of different treatment types across multi-site, long-term experiments [242]. In R, we used multi-level meta-analysis to test null hypotheses that mean effect sizes are equal to zero (*z*-test; *H_o_*: μ = 0; α = 0.05). Mixed-effect models accounted for variances associated with sampling error, the random effects of study site (i.e., between-study error), and repeated sampling over time at study sites (i.e., within study error) [85,243]. In addition, fixed effects of treatment and plant community type were coded as moderators. For each variable, we conducted three analyses: (1) a comparison of the three treatment types (plant communities pooled), (2) separate comparisons of plant community type for each treatment type, and (3) an overall analysis that included the entire dataset. Effect sizes were considered significant if 95% confidence intervals did not overlap zero [227,244,245]. Similarly, two effect sizes were considered significantly different if 95% confidence intervals did not overlap each other.

## Figures and Tables

**Figure 1 plants-09-01113-f001:**
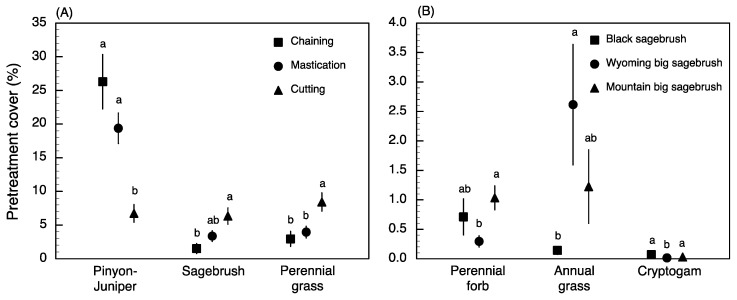
Mean (± SE) pretreatment cover showing differences among treatments (**A**) and plant community types (i.e., black sagebrush (*Artemisia nova*), Wyoming big sagebrush (*A. tridentata* ssp. *wyomingensis*), and mountain big sagebrush (*A. tridentata* ssp. *vaseyana*]) (**B**).

**Figure 2 plants-09-01113-f002:**
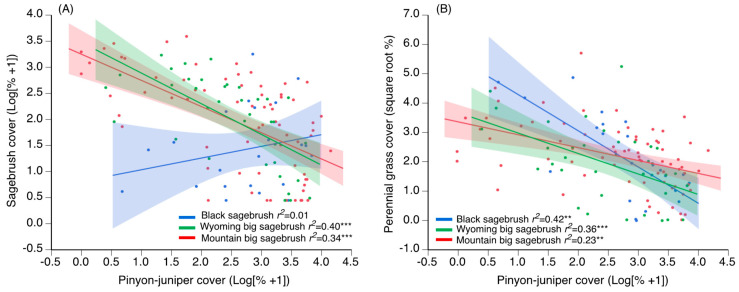
Prediction lines for linear relationships between pretreatment cover of pinyon–juniper and sagebrush (**A**) and between pinyon–juniper and perennial grass (**B**) in three sagebrush community types (i.e., black sagebrush (*Artemisia nova*; *n* = 43), Wyoming big sagebrush (*A. tridentata* ssp. *wyomingensis*; *n* = 63), and mountain big sagebrush (*A. tridentata* ssp. *vaseyana*; *n* = 128). Shaded regions are 95% confidence intervals. Asterisks indicate significance (*** *p* < 0.001, ** *p* < 0.01).

**Figure 3 plants-09-01113-f003:**
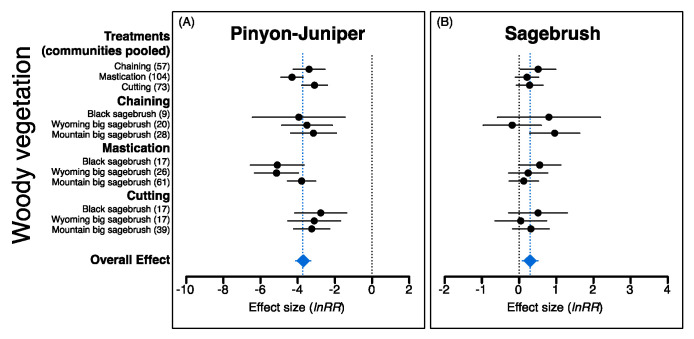
Mean effect sizes and 95% confidence intervals (*lnRR*; pre vs. posttreatment cover) for pinyon-juniper (**A**) and sagebrush (**B**) responses following pinyon–juniper reduction in three sagebrush community types (i.e., black sagebrush (*Artemisia nova*), Wyoming big sagebrush (*A. tridentata* ssp. *wyomingensis*), and mountain big sagebrush (*A. tridentata* ssp. *vaseyana*) when treatments are both pooled and separated (solid black circles). Overall effects (blue diamonds) indicate effect sizes for all observations; values in parentheses are the number of observations used to calculate effect sizes. Effect sizes are considered significantly different from one another if confidence intervals do not overlap.

**Figure 4 plants-09-01113-f004:**
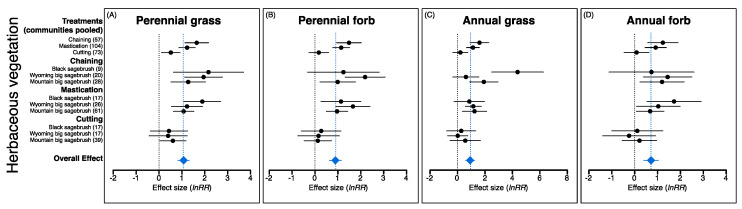
Mean effect sizes and 95% confidence intervals (*lnRR*; pre vs. posttreatment cover) for perennial grass (**A**), perennial forb (**B**), annual grass (**C**), and annual forb (**D**) vegetation responses following pinyon–juniper reduction in three sagebrush community types (i.e., black sagebrush (*Artemisia nova*), Wyoming big sagebrush (*A. tridentata* ssp. *wyomingensis*), and mountain big sagebrush (*A. tridentata* ssp. *vaseyana*)) when treatments are both pooled or separated (solid black circles). Overall effects (blue diamonds) indicate effect sizes for all observations; values in parentheses are the number of observations used to calculate effect sizes. Effect sizes are considered significantly different from one another if confidence intervals do not overlap.

**Figure 5 plants-09-01113-f005:**
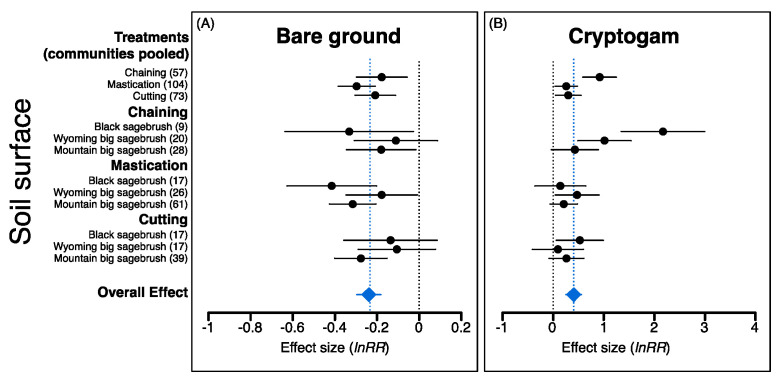
Mean effect sizes and 95% confidence intervals (*lnRR*; pre vs. posttreatment cover) for bare ground (**A**) and cryptogam (**B**) following pinyon–juniper reduction in three sagebrush community types (i.e., black sagebrush (*Artemisia nova*), Wyoming big sagebrush (*A. tridentata* ssp. *wyomingensis*), and mountain big sagebrush (*A. tridentata* ssp. *vaseyana*)) when treatments are both pooled or separated (solid black circles). Overall effects (blue diamonds) indicate effect sizes for all observations; values in parentheses are the number of observations used to calculate effect sizes. Effect sizes are considered significantly different from one another if confidence intervals do not overlap.

**Figure 6 plants-09-01113-f006:**
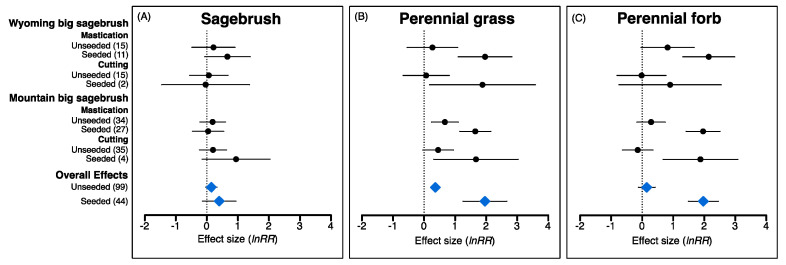
Mean effect sizes and 95% confidence intervals (*lnRR*; pre vs. posttreatment cover) for sagebrush (**A**), perennial grass (**B**), and perennial forb (**C**) responses in three sagebrush community types following pinyon–juniper reduction by mastication and cutting treatments (solid black circles) for unseeded and seeded sites. Overall effects (blue diamonds) indicate effect sizes for all observations; values in parentheses are the number of observations used to calculate effect sizes. Effect sizes are considered significantly different from one another if confidence intervals do not overlap.

**Figure 7 plants-09-01113-f007:**
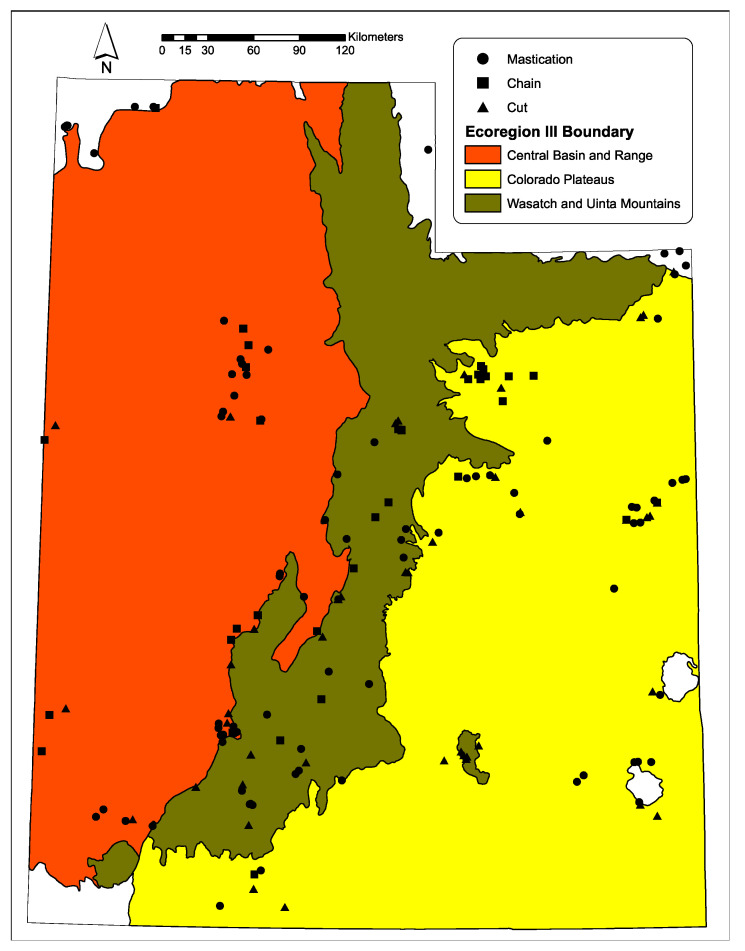
Map of pinyon–juniper reduction study sites in Utah.

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
