# Peer review of "Understory Vegetation Change Following Woodland Reduction Varies by Plant Community Type and Seeding Status: A Region-Wide Assessment of Ecological Benefits and Risks"

_plants, 2020, doi:10.3390/plants9091113_

Round 1

Reviewer 1 Report

I appreciate the authors' effort to complete a thorough and rigorous revision of their manuscript. The revision addresses all of my concerns, and the paper is a really nice synthesis of complex results.

Minor typo in abstract:

There is a close-parentheses missing on Line 23 after "treatment cover".

Author Response

Minor typo in abstract:

There is a close-parentheses missing on Line 23 after "treatment cover".

This typographic error has been fixed.

Reviewer 2 Report

Dear authors,

I accept the logic behind the paper but I am not convinced after reading it. Please remember that there is broad forest science saying that forests increase ecosystem services (opposite what you are saying). Forests protect water and soil, mitigate negative consequences of climate change and so on…Sustainable Forest Management include even prescribed fires, small water retention not mention about such externalities like tourism, green funerals, recreation and health healing or CO2 sequestration. Many openings in forests (clear cuttings and young plantations, fields designed for game feeding etc. supply room for diverse flora and fauna. Therefore forests are considered to be the most diverse terestrial, natural ecosystems. This aspect could be better discussed to justify putting so much efforts to stop natural succession. But this could be the subject for the next research.

In this paper I would suggest to replace "spp." with "spp." across the paper eg. Lines 15-16, 66…

L22 both signs “(“ and “[“ start but not end?

L 92 erase extra space after "understood"

Author Response

I accept the logic behind the paper but I am not convinced after reading it. Please remember that there is broad forest science saying that forests increase ecosystem services (opposite what you are saying). Forests protect water and soil, mitigate negative consequences of climate change and so on…Sustainable Forest Management include even prescribed fires, small water retention not mention about such externalities like tourism, green funerals, recreation and health healing or CO2 sequestration. Many openings in forests (clear cuttings and young plantations, fields designed for game feeding etc. supply room for diverse flora and fauna. Therefore forests are considered to be the most diverse terestrial, natural ecosystems. This aspect could be better discussed to justify putting so much efforts to stop natural succession. But this could be the subject for the next research.

Indeed, forests provide innumerable ecosystem services. Tree encroachment, expansion, and infilling, however, are a major global concern. Natural succession in many situations has been co-opted by human influences, including exotic plant invasion, resulting in novel ecosystem behaviors and diminished ecosystem services. The end result, as illustrated in our assessment in western U.S. shrub ecosystems, is a critical situation that required human intervention. We avoided being a proponent of reduction, but explained why conifer reduction projects have been initiated. Our emphasis was clearly to illustrate the outcomes of reversing conifer encroachment. We do this by weighing benefits and risks. The benefits methods to maintain heterogeneity, with mixed conifer shrub ecotones, are an the topic of future studies. In fact, ancient forests are not routinely treated, but preserved due to long-term succession, where trees have been present for prior to European settlement of western North America. Thus, the characterization of site history is a major research thrust in this region (as we indicate).

In this paper I would suggest to replace "spp." with "spp." across the paper eg. Lines 15-16, 66…

Your suggested edits were made.

L22 both signs “(“ and “[“ start but not end?

A bracket has been added to this line to close the equation.

L 92 erase extra space after "understood"

The appearance of an extra space where you indicated is actually caused by document formatting (with left and right justification).

Reviewer 3 Report

Thanks very much for the authors’ efforts in revising this manuscript by redoing the meta-analysis and rewriting Introduction, Results and Discussion. Most of my concerns have been addressed appropriately and I felt that the quality and rigorous of this manuscript has been largely improved. I only have a few minor issues that need to be considered before acceptance for publication.

  1. In the calculation of LnRR, the authors used measures of pre-treatment on the same site of removal as the control to evaluate the outcomes of woody removal. Generally, woody removal studies used additional woody plant intact sites as the control to avoid any confounding effect due to the temporal variations in ecological responses after woody removal. Therefore, it would be better to add a sentence in Method to clarify that using pre-treatment as control in this study is reasonable.

  1. L177-179, why mountain big sagebrush exhibits greater resilience to disturbance? Any literatures to support it?

  1. Fig.1 I would suggest adding different low case letters (a, b, c etc) to indicate the significant difference among treatment types or plant communities. For example, the significant difference among plant communities on cryptogam is not clear based on the current figure.

  1. The coloured background of Fig. 1, 3, 4, 5 blurs with the blue colour used for the overall effects. I suggest changing these figure background to white or transparent.

  1. The subtitle of 3.3 is a bit of hard to understand. What’s “the patterns in understory resilience”? Please revise it.

  1. L386-387. I was bit of confused. Seeding is a post-treatment management after woody removal, why it relates to pre-treatment abundance of perennial understorey? Similar issue in L194-197. I suggest adding a sentence to clarify the seeding at L386-387 or at where it first appears in the manuscript.

Author Response

Thanks very much for the authors’ efforts in revising this manuscript by redoing the meta-analysis and rewriting Introduction, Results and Discussion. Most of my concerns have been addressed appropriately and I felt that the quality and rigorous of this manuscript has been largely improved. I only have a few minor issues that need to be considered before acceptance for publication.

In the calculation of LnRR, the authors used measures of pre-treatment on the same site of removal as the control to evaluate the outcomes of woody removal. Generally, woody removal studies used additional woody plant intact sites as the control to avoid any confounding effect due to the temporal variations in ecological responses after woody removal. Therefore, it would be better to add a sentence in Method to clarify that using pre-treatment as control in this study is reasonable.

We acknowledged the limitations of our study on line 536, prior to justifying the use of LnRR (pre- vs. post-treatment). In addition, we added a statement indicating that our approach was necessary because control areas were not available for analysis, as is typical in woody plant reduction studies.

L177-179, why mountain big sagebrush exhibits greater resilience to disturbance? Any literatures to support it?

We added a few citations to support this hypothesis.

Fig.1 I would suggest adding different low case letters (a, b, c etc) to indicate the significant difference among treatment types or plant communities. For example, the significant difference among plant communities on cryptogam is not clear based on the current figure.

Lower-case letters from mean separation tests were added to this figure.

The coloured background of Fig. 1, 3, 4, 5 blurs with the blue colour used for the overall effects. I suggest changing these figure background to white or transparent.

The background on all figures were changed to white.

The subtitle of 3.3 is a bit of hard to understand. What’s “the patterns in understory resilience”? Please revise it.

We changed the term patterns to "differences."

L386-387. I was bit of confused. Seeding is a post-treatment management after woody removal, why it relates to pre-treatment abundance of perennial understorey? Similar issue in L194-197. I suggest adding a sentence to clarify the seeding at L386-387 or at where it first appears in the manuscript.

Yes, you are correct-- seeding occurs after treatment. Our goal was to illustrate  that because pretreatment conditions varied, it dictated whether a site is seeded or not. We follow up on this point on lines 386-389, and to make this connection clearer, we reference figure 1 on lines 386-389.